# Learning with Weak Annotations for Robust Maritime Obstacle Detection

**DOI:** 10.3390/s22239139

**Published:** 2022-11-25

**Authors:** Lojze Žust, Matej Kristan

**Affiliations:** Faculty of Computer and Information Science, University of Ljubljana, Večna pot 113, 1000 Ljubljana, Slovenia

**Keywords:** semantic segmentation, weak supervision, obstacle detection, maritime perception

## Abstract

Robust maritime obstacle detection is critical for safe navigation of autonomous boats and timely collision avoidance. The current state-of-the-art is based on deep segmentation networks trained on large datasets. However, per-pixel ground truth labeling of such datasets is labor-intensive and expensive. We propose a new scaffolding learning regime (SLR) that leverages weak annotations consisting of water edges, the horizon location, and obstacle bounding boxes to train segmentation-based obstacle detection networks, thereby reducing the required ground truth labeling effort by a factor of twenty. SLR trains an initial model from weak annotations and then alternates between re-estimating the segmentation pseudo-labels and improving the network parameters. Experiments show that maritime obstacle segmentation networks trained using SLR on weak annotations not only match but outperform the same networks trained with dense ground truth labels, which is a remarkable result. In addition to the increased accuracy, SLR also increases domain generalization and can be used for domain adaptation with a low manual annotation load. The SLR code and pre-trained models are freely available online.

## 1. Introduction

Autonomous boats have significant commercial and societal potential for transoceanic cargo shipping, passenger transport, hazardous area inspection, and environmental control. Their autonomy crucially relies on obstacle detection, which is particularly challenging in maritime environments such as coastal waters, marinas, urban canals, and rivers. This is because the appearance of the navigable surface (water) is dynamic, changes with the weather, and contains object reflections and glitter. Similarly, obstacles can be static (e.g., shore and piers) or dynamic (e.g., boats, swimmers, debris, buoys) and have a wide range of appearances.

The current state of the art in vision-based maritime obstacle detection [1] is based on the segmentation of captured images into water, obstacle, and sky classes. Unlike detection-based methods, segmentation methods can simultaneously detect static and dynamic obstacles, are more robust to the diverse appearance of obstacle types, and directly detect the navigable area (water). However, their performance depends on the availability of large per-pixel segmented training datasets [2].

Manual segmentation of training sets is time-consuming, costly, and error-prone. For example, manual segmentation of a typical maritime image takes approximately 20 min [2]. In a related field of autonomous vehicles, significant efforts have thus been invested in semi-automatic annotation [3,4], domain transfer [5,6], and semi-supervised learning [7,8] to reduce the required labeling effort. Alternatively, weak supervision methods [9,10] aim to achieve this by simplifying annotations and harnessing prior knowledge about image and data structure. Unfortunately, these methods are largely inapplicable to the maritime domain due to their reliance on overly general annotations such as image-level labels or object bounding boxes, which are unable to capture all the requirements for robust maritime perception. Instead, the annotation effort may be reduced in a more principled way by considering the specifics of downstream applications. Thus far, this direction has not yet been explored in the domain of maritime obstacle detection.

We observe that segmentation errors in different semantic regions may have profoundly different consequences for maritime navigation. Figure 1 visualizes several of these cases. Detecting the boundary between water and static obstacles (i.e., the water edge) is crucial for collision prevention, while accurate segmentation of the shore–sky boundary is irrelevant for obstacle avoidance. Similarly, misclassifying a few pixels on a floating obstacle will not cause a collision, but falsely classifying isolated patches of water pixels as an obstacle adversely affects control, causing frequent and unnecessary stops. Recent maritime obstacle detection benchmarks [1] also reflect these navigation-specific segmentation requirements by defining performance evaluation measures that evaluate the performance in terms of dynamic obstacle detection and water–edge estimation accuracy, while ignoring the segmentation accuracy beyond the water edge.

Taking into account the aforementioned requirements of the maritime obstacle detection task, we propose a new scaffolding learning regime (SLR), which is our main contribution. SLR avoids the need for densely labeled ground truth for training maritime obstacle detection networks and instead relies only on weak obstacle-oriented annotations (Figure 1) consisting of water–edge poly-lines, bounding boxes denoting dynamic obstacles, and the horizon location estimated from the on-board inertial measurement unit (IMU) data. At a high level, SLR works with EM-like steps, alternating between estimating the unknown segmentation labels and improving the network parameters. First, by considering the domain constraints, we can construct partial segmentation labels (i.e., not all pixels are labeled) from weak annotations. A segmentation network is then trained on the partial labels to learn domain-specific feature extraction. During this initialization step, we also use additional bounding-box-based training objectives to learn the segmentation of foreground dynamic obstacles.

In turn, the trained network is used to estimate the labels in the unknown regions of the image. While the predictions of such a network cannot be expected to reach the desired robustness due to the complex interplay of multiple training objectives, the encoder must learn powerful domain-specific features to satisfy all the objectives. We therefore use a feature clustering approach to estimate the most likely labels for unlabeled pixels in the partial segmentation labels. Finally, the network is fine-tuned using the newly estimated pseudo-labels.

Extensive evaluation on maritime obstacle detection by segmentation [1] shows that models trained using SLR outperform models classically trained on full dense annotations, which is a remarkable result. In fact, the new training regime increases robustness to false-positive detections and improves the generalization capabilities of the trained networks, while reducing the time required for ground truth annotation by orders of magnitude. To the best of our knowledge, this is the first method for training obstacle detection from weak annotations in the marine domain which surpasses fully supervised training from dense labels. Furthermore, SLR makes minimal assumptions on the network architecture and can thus be applied to most of the existing segmentation networks. Additionally, it is only used during training and thus does not impact network inference characteristics such as speed or hardware requirements.

Preliminary results of our approach were presented in a conference paper [11]. This paper goes beyond the preliminary work in several ways. We introduce an additional auxiliary object loss ($L_{\mathrm{aux}}$), which uses segmentation priors, estimated from bounding boxes, to supervise dynamic obstacle segmentation. Additionally, we reformulate the retraining step as fine-tuning, starting training from pre-trained weights of the warm-up stage. Both changes lead to substantial performance improvements. The experimental analysis is also significantly extended with a cross-domain generalization analysis (Section 4.5), an annotation efficiency study including a comparison with a recent state-of-the-art annotation–reduction approach (i.e., semi-supervised learning, Section 4.4), experiments and discussion about soft labels (Section 4.8), as well as qualitative analysis on a wide range of maritime images (Section 4.9).

## 2. Related Work

In the following, we overview the related work in maritime obstacle detection (Section 2.1) and label-efficient training for deep segmentation models (Section 2.2).

### 2.1. Maritime Obstacle Detection

Early approaches for obstacle detection in the marine domain include handcrafted methods such as background subtraction [12], stereo reconstruction [13], and statistical semantic segmentation methods with color and texture features [14,15]. However, due to the challenging dynamics of the water surface, these methods struggle in challenging scenes and tend to produce a large amount of FP detections.

For this reason, similarly to autonomous ground vehicles (AGV), the perception in marine robotics has shifted towards deep learning methods in recent years, enabling the learning of rich visual features specialized for the target task. Early works [16,17,18] were based on the direct application of classical deep general object detection methods [19] or their specializations to ship detection tasks [20]. Unfortunately, these methods only detect obstacles that can be pre-trained and are in the form of compact well-defined objects, while they are unable to address static obstacles such as piers, shorelines, and floating fences. Furthermore, the wide variety of obstacle appearances, which is specific to the often poorly structured maritime domain, prohibits training of the object-category-specific detectors used in these works.

To address these limitations, several works investigated deep-learning-based semantic segmentation as a possible alternative, posing the problem of obstacle detection as anomaly segmentation, where all obstacles (static and dynamic) are assigned to a single obstacle class. However, due to the specifics of the maritime domain (dynamics of the water surface, reflections), applying well-established semantic segmentation models from the AGV domain [21,22] to the maritime domain lacks the desired robustness [2,23], as it achieves high recall rates but low precision and struggles to deal with the dynamic appearance of the water surface.

For this reason, several networks designed specifically for the maritime domain have been recently proposed [24,25,26,27]. Networks [25,28] are based on the U-net architecture [29] and utilize various regularization and data augmentation techniques to increase their robustness. Recently, Ref. [26] introduced a novel architecture WaSR, which harnesses additional information from the on-board IMU measurements to estimate the horizon location prior, which is fused with the image features in the network decoder. Additionally, a water–separation loss is proposed to encourage learning a better separation of water and obstacle features in the model encoder. The approach greatly reduces the sensitivity of the network to changes in the water appearance, halving the number of FP detections, and is the current state-of-the-art on the USV obstacle detection benchmark MODS [1]. A maritime domain panoptic segmentation approach has also been proposed [27], enabling the distinction between obstacle instances, but has only been applied to a limited task of ship detection [30] from mostly static on-shore sequences.

As shown by [1], current state-of-the-art approaches still lack the desired robustness, especially within the most critical 15 m radius around the boat (i.e., the danger zone) and struggle with domain-specific artefacts such as waves, object reflections, and sun glitter. Further development of the field is also limited by the low availability of per-pixel annotated datasets [2,27,31], the annotation of which is costly and error-prone. Our work partly addresses this issue by proposing a method for learning maritime semantic segmentation networks from weak labels, thereby avoiding the need for per-pixel labeling and significantly reducing the labeling effort. In addition, the method improves the robustness of the learned networks by considering the downstream task during training.

### 2.2. Reducing the Annotation Effort

The development of deep segmentation models is hampered by their reliance on large amounts of painstakingly labeled training data. Recent annotation-efficient training methods thus aim to achieve a better trade-off between model performance and annotation effort. Semi-supervised methods [7,8,32,33] aim to harness additional unlabeled images to improve segmentation accuracy. Theoretically, this could also reduce the required labeling effort by reducing the amount of required labeled images. However, as the annotation effort is mainly reduced by labeling fewer images, such methods are more sensitive to labeling errors and can only capture a limited visual variation of open-ended classes such as obstacles.

Alternatively, weakly-supervised methods [34,35] reduce the annotation effort by using weaker forms of labels. Approaches such as expectation maximization [36], multiple instance learning [37], and self-supervised learning [38,39,40,41] have been considered to learn semantic segmentation of objects from image-level labels. These approaches can infer information about class appearances from the differences between images containing and images not containing objects of a certain class. As such, they are well-suited for natural images, with a relatively small number of objects per image and a large vocabulary of classes, where image-level labels are informative to the content of the image. However, for general scene parsing required for obstacle detection in autonomous vehicles and boats, where many objects are present in the scene at once, often including multiple instances of the same class (i.e., obstacles), such approaches do not perform well [10]. This is especially problematic in domains with a small number of classes such as anomaly segmentation in the marine domain (3 classes), where all classes are present in most of the scenes and thus image-level labels provide almost no discriminative information.

Other approaches such as scribbles [42,43,44] and point annotations [3,45] have also been considered for this task. These provide the information for the most representative pixels of an object but fail to capture the extent or boundaries of objects, which are very important for boat navigation. Bounding boxes are much more suitable for this task, as they provide information both about the location of an object as well as weak constraints on its boundary. Due to these informative cues, ease of annotation, and their ubiquitous presence, bounding boxes have been extensively studied as weak constraints in many segmentation tasks, including semantic segmentation [46,47], instance segmentation [48,49], and video object segmentation [50], where they have significantly reduced the performance gap to fully supervised methods over the last few years. However, these approaches mainly focus on the segmentation of well-defined foreground objects (e.g., dynamic obstacles) but are not designed to learn boundaries between background classes (e.g., static obstacles), which are required for general scene parsing and obstacle detection in AGVs or USVs.

Instead, general scene parsing must account both for the segmentation of foreground objects as well as background classes, thus a combination of different weak-annotation types may be more appropriate. The method introduced by [9] uses bounding boxes to supervise the learning of foreground classes, while image-level labels are used to supervise the learning of the background class, which is possible due to a large vocabulary of background classes. This, however, is not the case in the maritime domain, where there are only three classes. Furthermore, from the perspective of maritime obstacle detection, a single aspect of background segmentation is the most important—the boundaries of the water surface. Instead of image-level labels, we thus propose the use of water–edge annotations for supervising the background segmentation, thus focusing the annotation effort on this crucial information, at a comparable effort to less focused image-level labels.

## 3. Learning Segmentation by Scaffolding

We now introduce the new learning approach for maritime segmentation networks, which we call Scaffolding Learning Regime (SLR). SLR gradually improves the trained model by iterating between improving the network parameters using per-pixel (pseudo-) labels and re-estimating the pseudo-labels using the learned network. This process is composed of three steps shown in Figure 2. In the first step (Section 3.1), the network is trained using (1) partial labels, which are constructed from weak annotations and domain constraints and (2) additional weak objectives for learning the segmentation of dynamic obstacles. In the second step (Section 3.2), the learned network is used to estimate the labels of unlabeled regions in partial labels. Finally (Section 3.3), the network is fine-tuned using the estimated pseudo-labels. These three steps are explained in more detail below.

### 3.1. Feature Warm-Up

The purpose of the feature warm-up step is to learn domain-specific encoder features and initial segmentation predictions. This is achieved through weakly supervised training. By combining domain knowledge and weak annotations, we can label specific regions of an input image $I\in R^{W\times H\times 3}$ with high confidence, while leaving other regions unlabeled, resulting in partial labels $\overset{˚}{Y}\in {\left[0,1\right]}^{W\times H\times 3}$ (Section 3.1.1), which can be used to supervise the model training (Section 3.1.2).

#### 3.1.1. Partial Labels from Weak Annotations

To generate per-pixel partial labels $\overset{˚}{Y}=\left({\overset{˚}{Y}}_{w},{\overset{˚}{Y}}_{s},{\overset{˚}{Y}}_{o}\right)$ for the water, sky, and obstacle classes, respectively, we introduce domain-specific constraints extrapolated from weak annotations and the horizon location estimated from the IMU measurements (following [26]), as shown in Figure 3. The estimated horizon divides the image into two groups: regions above the horizon ($H^{↑}$) and regions below the horizon ($H^{↓}$). Similarly, water–edge annotations define the sets $W^{↑}$ and $W^{↓}$ for the regions above and below, respectively, and bounding boxes define the set O of rectangular regions that tightly enclose dynamic obstacles.

Using this notation, we define class-constrained regions $C_{c}$, where the class *c* cannot appear. Namely, water pixels cannot appear above the horizon or water edge ($C_{w}=H^{↑}\;\cup \;W^{↑}$), sky pixels cannot appear below the horizon or water edge ($C_{s}=H^{↓}\;\cup \;W^{↓}$) and obstacle pixels cannot appear outside the object bounding boxes except above the water edge ($C_{o}=O^{C}\backslash W^{↑}$). We can thus set the probability for the class *c* of a pixel *i* to 0 within the respective restricted region. In certain regions, these constraints lead to unambiguous labels with only one class remaining. In general, the probability of the class *c* at pixel position *i* is defined as
(1)$${\overset{˚}{Y}}_{c}^{i}=\left\{\begin{array}{cc} 1 & \mathrm{if}\;i\notin C_{c}\wedge i\in {⋂}_{k\neq c}C_{k},\\ 0 & \mathrm{otherwise}. \end{array}\right.$$

To account for the lack of unique labels on some pixels, we introduce pixel-wise training weights $w_{i}$. In particular, the weights are set to $w_{i}=1$ for pixels with unambiguous labels and to $w_{i}=0$ elsewhere. This means that the latter pixels are effectively ignored during training.

From the water–edge annotations, we can also infer that a pixel *i*, located immediately above the water edge, must belong to the obstacle class (${\overset{˚}{Y}}_{o}^{i}=1$) by definition. However, since the height of the static obstacle is unknown, we employ a heuristic approach and only label a pixel as an obstacle if its distance $d_{i}$ from the water edge is below a threshold θ, i.e.,
(2)$${\overset{˚}{Y}}_{o}^{i\in W^{↑}}=\left\{\begin{array}{cc} 1 & \mathrm{if}\;d_{i}<\theta ,\\ 0 & \mathrm{otherwise}. \end{array}\right.$$

We furthermore adjust its weight to reflect the increase of label uncertainty with distance, i.e., $w_{i}=\exp\left(-\alpha d_{i}\right)$, where $\alpha =-ln(\omega_{\min})/\theta$ is defined such that all weights lower than a small value $\omega_{\min}$ are set to zero.

#### 3.1.2. Training

The network is trained with a weighted focal loss [51] $L_{\mathrm{foc}}$ on the partial labels $\overset{˚}{Y}$ and their corresponding weights $w_{i}$. However, further training signals can be derived for the unlabelled regions corresponding to the dynamic obstacles.

In particular, we use several loss functions inspired by instance segmentation literature [47,49]. We leverage the bounding box annotations with a projection loss $L_{\mathrm{proj}}$ [49]. The projection loss provides a weak constraint on the segmentation of an obstacle and forces the horizontal and vertical projections of the segmentation mask to coincide with the edges of the bounding box. Further regularization is provided by using a pairwise loss $L_{\mathrm{pair}}$ [49]. This loss promotes equal labels for visually similar neighboring pixels. We adapt the pairwise loss term to a multiclass environment and apply it to the entire image so that it monitors both dynamic and static obstacle segmentation.

Finally, we estimate a prior for the segmentation mask for each dynamic obstacle using a pre-trained class-agnostic deep grab-cut segmentation network [52]. In turn, a focal loss between the predicted object segmentation and the estimated prior segmentation $L_{\mathrm{aux}}$ is used as an auxiliary source of obstacle segmentation supervision.

The final loss is thus composed of global and dynamic obstacle losses:(3)$$L_{\mathrm{wrm}}=L_{\mathrm{foc}}\left(\overset{˚}{Y};w\right)+L_{\mathrm{pair}}+\sum_{n=1:N}\left(L_{\mathrm{proj}}^{\left(n\right)}+L_{\mathrm{aux}}^{\left(n\right)}\right),$$
where *N* is the number of annotated dynamic obstacles.

### 3.2. Estimating Pseudo Labels from Features

The model learned in the warm-up phase (Section 3.1) cannot be expected to produce robust predictions due to a complex combination of training objectives used during training. However, to address all training objectives simultaneously, the encoder must learn strong domain-specific features. We thus estimate the labels of unlabeled regions of the partial labels $\overset{˚}{Y}$ based on feature clustering in the learned feature space producing dense pseudo labels $\tilde{Y}$. This process is based on the assumption that pixels corresponding to the same semantic class cluster together in the learned feature space.

We first correct the model predictions with constraints derived from weak annotations. Let $\hat{Y}=\left({\hat{Y}}_{w},{\hat{Y}}_{s},{\hat{Y}}_{o}\right)\in {\left[0,1\right]}^{W\times H\times 3}$ be the model predictions (probabilities) for the water, sky, and obstacle class, and $F\in R^{W\times H\times C}$ be the feature maps produced by the encoder for an input image I. We define r(·) as a function that corrects (i.e., constrains) the labels according to the domain constraints—the probability for class *c* is set to 0 in restricted locations *i* as in Section 3.1.1, i.e., $r\left({\hat{Y}}_{c}^{i}\right)=0;\forall i\in C_{c}$

From the constrained predictions $R=r(\hat{Y})$, class prototypes are constructed. A class prototype $p_{c}$ is a single feature vector describing the class *c* and is computed as a masked average pooling over the features
(4)$$p_{c}=\frac{\sum_{i\in I}R_{c}^{i}F^{i}}{\sum_{i\in I}R_{c}^{i}},$$
where $F^{i}$ and $R_{c}^{i}$ denote the features and constrained probabilities of the class *c* at an image location *i*. Because dynamic obstacle appearance might vary greatly across instances, we construct individual prototypes $p_{d_{1}},\ldots ,p_{d_{N}}$ for each of the dynamic obstacles in the image (from features within the obstacle bounding box), and a separate prototype $p_{\mathrm{st}}$ for all the remaining static obstacles. Two prototypes, $p_{w}$ and $p_{s}$, are extracted for the water and the sky class, respectively.

The probability of a pixel belonging to a specific class is reflected in the similarity between the pixel and the respective class prototype in the feature space. High similarity relative to the other prototypes indicates high class probability, while low relative similarity indicates low probability. To implement this, we first compute the feature similarity with the constructed prototypes at every image location to produce similarity maps S for each of the prototypes. The similarity map $S_{c}$ for class *c* at image location *i* is computed by cosine similarity
(5)$$S_{c}^{i}=\frac{F^{i}\cdot p_{c}}{∥F^{i}∥∥p_{c}∥}.$$

Due to the introduction of multiple prototypes for the obstacle class, we obtain multiple obstacle similarity maps, which need to be merged into a single obstacle similarity map $S_{o}$ as follows: similarity maps for each of the dynamic obstacles ($S_{d_{1}},\ldots ,S_{d_{N}}$) are applied inside their respective obstacle bounding boxes, and the static obstacle similarity map $S_{\mathrm{st}}$ is used elsewhere. In regions where annotations of multiple dynamic obstacles overlap, the maximum of their respective similarity values is used.

We can then apply a softmax function over the class similarities to obtain the class probability distribution at each image position *i*, i.e.,
(6)$${\tilde{P}}_{c}^{i}=\frac{\exp(\beta \;S_{c}^{i})}{\sum_{k}\exp(\beta \;S_{k}^{i})},$$
where $S=\left(S_{w},S_{s},S_{o}\right)$ are the class similarity maps, and β is a scaling hyper-parameter. The resulting probabilities are corrected using r(·) to agree with the weak annotation constraints.

The estimated probabilities are used as soft pseudo-labels to fine-tune the model. Additionally, since we have already constructed the partial labels (Section 3.1.1), we only need to apply the estimated probabilities to unlabeled (i.e., unknown) regions in the partial labels $\overset{˚}{Y}$. The final pseudo labels can thus be expressed as
(7)$${\tilde{Y}}^{i}=\left\{\begin{array}{cc} r({\tilde{P}}^{i}) & \mathrm{if}\;i\;\mathrm{not}\;\mathrm{labeled}\;\mathrm{in}\;\overset{˚}{Y},\\ {\overset{˚}{Y}}^{i} & \mathrm{otherwise}. \end{array}\right.$$

Finally, the weights of pixels whose probability is estimated by ${\tilde{P}}^{i}$ are set to a constant weight $w_{i}=\omega_{R}<<1$ to reflect that the estimated labels are less certain than the partial labels derived directly from weak annotations.

### 3.3. Fine-Tuning with Dense Pseudo Labels

In the final step, the model trained in the warm-up stage is fine-tuned by optimizing a weighted focal loss between the predicted labels and the re-estimated dense pseudo labels and a global pairwise loss, i.e.,
(8)$$L_{\mathrm{finetune}}=L_{\mathrm{foc}}\left(\tilde{Y}\right)+L_{\mathrm{pair}}.$$

## 4. Results

A battery of experiments was conducted to probe the learning capabilities of SLR in the context of maritime obstacle detection. In all experiments, the focal loss parameter is set to γ=2 and the remaining SLR hyper-parameters to θ=11, $\omega_{\min}=0.005$, β=20 and $\omega_{R}=0.5$. The number of training epochs in the warm-up and fine-tuning phases is 25 and 50, respectively. Unless stated otherwise, we stop the training after a single SLR iteration (i.e., pseudo-label estimation and model fine-tuning).

SLR is demonstrated on the state-of-the-art maritime obstacle detection network WaSR [26], which employs a ResNet-101 backbone as the encoder. In the warm-up phase, the encoder is initialized from pre-trained weights on Imagenet, while the remaining network weights are initialized randomly. Features from the penultimate (third) encoder residual block are used in the pseudo-label estimation phase after warm-up (Section 3.2). The water separation loss with weight $\lambda_{\mathrm{ws}}=0.01$ from WaSR is added to the losses in the fine-tuning step (Section 3.3).

Since our goal is to compare SLR with classical learning, we have tried to minimize the number of changes compared to WaSR to allow a fair comparison. Thus, following [2], all networks are trained with RMSProp optimizer with momentum 0.9, initial learning rate $10^{-6}$, standard polynomial reduction decay 0.9 and a batch size of 12. Random image augmentations including color transformations, horizontal flipping, scaling, and rotation are applied to training images.

### 4.1. Evaluation Protocol

The standard obstacle detection evaluation protocol from the MODS [1] benchmark is used. The networks are trained on MaSTr1325 [2], which contains 1325 diverse, fully per-pixel labeled images captured from unmanned surface vehicles (USV). To evaluate SLR, we additionally annotated the images with water edges and object bounding boxes. The models are evaluated on the test dataset from MODS, which contains approximately 100 sequences annotated by bounding boxes and water edges, using the detection-oriented evaluation protocol [1]. Static obstacle detection is evaluated by water–edge detection robustness ($\mu_{R}$), which measures the proportion of the water–edge boundary that has been correctly identified (i.e., the boundary is detected within a set threshold distance). Dynamic obstacle detection is evaluated by precision (Pr), recall (Re), and F1 measure. It is evaluated over the entire navigable surface and separately within a 15 m *danger zone* from the USV (F1${}_{D}$), where the obstacle detection performance is most critical for immediate collision prevention.

### 4.2. Comparison with Full Supervision

SLR was evaluated by training two of the top models from the recent MODS [1] benchmark, using the weak annotations from MaSTr1325 [2] and evaluating on the MODS test set. In addition, we trained several top-performing models from the same benchmark (RefineNet [53], DeepLabV3 [54], BiSeNet [55], and our re-implementation of WaSR [26]) using the dense labels to form a strong baseline. In the following, ${\left(\cdot \right)}_{\mathrm{SLR}}$ denotes the networks trained with SLR.

Results in Table 1 show that, remarkably, both WaSR${}_{\mathrm{SLR}}$ and DeepLabV3${}_{\mathrm{SLR}}$ outperform their classically-trained counterparts DeepLabV3 and WaSR, despite using considerably simpler weak annotations. DeepLabV3${}_{\mathrm{SLR}}$ outperforms its counterpart by 5.8 and 59.0 percentage points overall and inside the danger zone, respectively. In the case of WaSR, SLR boosts performance by 1.4 (overall) and 5.1 (danger zone) percentage points and sets a new state-of-the-art on the MODS benchmark. We observe that SLR consistently decreases false-positive detections and increases precision while preserving a high recall (qualitative results in Section 4.9). We speculate that this might be due to detection-based training objectives in SLR, which better reflect the downstream task requirements compared to the standard pixel-based segmentation losses.

### 4.3. Segmentation Quality

Since the test set annotations in MODS [1] do not enable segmentation accuracy analysis, we split the MaSTr1325 dataset into training (70%) and test (30%) sets. The standard WaSR and WaSR${}_{\mathrm{SLR}}$ are trained on the thus obtained training set and evaluated in terms of IoU on the segmentation ground truth of the test set.

Results in Table 2 show that the WaSR${}_{\mathrm{SLR}}$ segmentation accuracy closely matches that of WaSR with merely a 1.2 decrease in mIoU. This decrease can be attributed to slightly over-segmented objects, missed details, and thin structures in the sky, which are labeled as obstacles, but are not important from the obstacle detection perspective (Figure 4).

### 4.4. Comparison with Semi-Supervised Learning

SLR can be considered a weakly-supervised learning method, in which the segmentation supervision comes from weak obstacle-oriented annotations. Semi-supervised methods, on the other hand, use a small set of fully-labeled images and a larger set of unlabeled images in training. In both cases, the aim is to reduce the required annotation effort. We thus compare SLR with the recent semi-supervised state-of-the-art method ATSO [7] in terms of annotation efficiency. Table 3 compares the two methods and the classical fully-supervised training scheme with varying percentages of the images in the training set.

According to [2], manual segmentation of a MaSTr1325 image takes approximately 20 min. Since weak annotations take 1 min per image, we can estimate that annotation of all images with weak annotations is approximately 5% of the effort required to manually segment all images. We thus first analyze WaSR performance when trained with only 5% of all fully-segmented images (selected at random). Compared to using all training images, the performance in terms of the F1 detection measure substantially drops by almost 10 percentage points overall and 19.1 percentage points within the danger zone. Applying the semi-supervised approach ATSO with 5% of annotated (and 95% non-annotated) training images significantly improves the performance, particularly within the danger zone (by 18 percentage points). Nevertheless, SLR by far outperforms both the fully-supervised approach and ATSO and achieves nearly five percentage points overall and an over six percentage point improvement within the danger zone over ATSO. Even when increasing the annotation effort to 2× that of used with SLR by using 10% of annotations, ATSO still falls short by over three percentage points overall and almost seven percentage points within the danger zone. The results indicate that SLR and the obstacle-oriented reduction of labels to weak annotations are much more efficient than the general reduction of decreasing the number of labeled images.

For reference, we also evaluate the segmentation performance of the validation set of MaSTr1325. Fully supervised training using all images with ground truth segmentation labels achieves the best mIoU. Reducing the training images to match the annotation effort of SLR results in 3.5 drop in mIoU. This drop is slightly smaller when using ATSO (2.2 points) and smaller still when using SLR (1.2 points). The segmentation accuracy of ATSO matches that of SLR only when using twice as much manual annotation effort. These results suggest that SLR could potentially be also useful as a method for segmentation ground truth acquisition at a very low manual annotation effort.

### 4.5. Cross-Domain Generalization

MaSTr1325 [2] and MODS [1] both contain images acquired from the perspective of a small USV. To explore the cross-domain generalization advantages of SLR, we thus perform several experiments by evaluating on the Singapore Marine Dataset (SMD) [30]. This dataset presents a different domain than the MaSTr training set and contains video sequences mostly acquired from on-shore vantage points. The objects are annotated with weak annotations, and we use the horizon annotations and the training/test split from [26].

In the first experiment, we evaluate WaSR trained on MaSTr1325 and test it on the SMD test set. Table 4 shows that SLR outperforms training on segmentation ground truth by 5.1 percentage points, which indicates that better generalization capabilities are obtained purely from the proposed training regime.

We next consider the performance in the context of domain adaptation with MaSTr1325 and SMD training sets used in training and SMD test set for evaluation. We compare SLR with the recent state-of-the-art domain adaptation method FDA [6]. WaSR trained with FDA performs worse (19 percentage points drop) than WaSR trained only on the MaSTr1325 segmentation ground truth. However, using SLR to fine-tune the model on the SMD training set outperforms the latter by nearly 15 percentage points. This implies that SLR bears a strong potential in domain generalization as well as domain adaptation problems.

### 4.6. Ablation Study

To expose the contributions of individual components of SLR, a series of experiments with different components turned off was carried out. Results of training on MaSTr1325 and testing on MODS are reported in Table 5.

The most basic model that uses only the feature warm-up step (Section 3.1) without fine-tuning and without the static obstacle labels above the water edge (Section 3.1.1) results in a 7.2 and 38.8 percentage points drop overall and within the danger zone, respectively, compared to the full SLR. Detailed inspection showed that this is mainly due to an increased number of false-positive detections (see Figure 5). Adding the static obstacle labels in the warm-up step considerably improves the performance of the basic model (by 16.1 percentage points within the danger zone). A further boost of 13.9 percentage points within the danger zone is obtained by enabling the fine-tuning step (Section 3.3) and learning from predictions of the warmed-up model.

Applying the label constraints to predictions (r(·) from Section 3.2) leads to a further 4.4 percentage points improvement within the danger zone, while using only the re-estimation of pseudo labels with feature clustering (Section 3.2) leads to a 4.8 percentage points improvement. Combining both label constraints and pseudo-label re-estimation results in 3.7 (overall) and 6.5 (danger zone) percentage points improvements over the model without the predictions refinement step. Finally, using the auxiliary per-object segmentation loss ($L_{\mathrm{aux}}$) during warm-up further improves the performance by 0.7 and 2.3 points, respectively. The mIoU (Table 5) between the predicted segmentation masks and the ground truth labels on the validation set shows that each SLR module consistently contributes to learning the unknown underlying segmentation.

### 4.7. Influence of SLR Iterations

The number of SLR iterations composed of the pseudo-labels estimation (Section 3.2) and network fine-tuning (Section 3.3) steps is currently set to 1 in SLR. Table 6 reports results with increasing the number of iterations. For reference, performance without the fine-tuning step is reported as well. Results show that a single fine-tuning step significantly boosts the performance compared to not using it; however, results do not improve with further iterations. We thus conclude that a single fine-tuning step in SLR is sufficient.

### 4.8. Comparison with Label Smoothing

The pseudo-label estimation step (Section 3.2) produces soft labels, which means that the distribution over labels at a given pixel is not collapsed to a single mode. We, therefore, tested the hypothesis that the observed improvements might actually come from the smoothed training labels, an effect reported in literature [56].

We implemented a smoothing model that applies per-pixel and cross-pixel label smoothing as follows. Let $p_{i}$ be the label distribution at a pixel *i*. The new label distribution is given as $p_{i}^{\prime }=p_{i}\left(1-\alpha \right)+\frac{\alpha }{N}$, where α is the extent of smoothing and *N* is the number of label classes. The obtained label mask can be further spatially smoothed by a 2D Gaussian filter with size parameter σ. This model is applied to the one-hot ground truth segmentation labels in MaSTr1325.

Results of exhaustive search over (α,σ) are shown in Figure 6. As predicted by [56], both types of label smoothing do improve the performance of the classically trained WaSR, particularly within the danger zone, albeit often at a slight overall performance decrease. Nevertheless, SLR training considerably outperforms all label smoothing combinations in both measures, despite using only weak annotations.

### 4.9. Qualitative Analysis

Figure 7 visualizes the performance of WaSR and WaSR${}_{\mathrm{SLR}}$ on the MODS [1] dataset. Observe that SLR training reduces false-positive detections on wakes, sun glitter, and reflections. Reduction of these is crucial for practical USV application since false positives result in frequent unnecessary slow-downs and effectively render the perception useless for autonomous navigation. Note that, while segmentation and thus obstacle detection substantially improve in regions important for navigation, structures such as masts are missed. However, these are in the areas irrelevant for navigation and not accounted for by supervision signals in SLR.

We next downloaded and captured several diverse maritime photographs. These images were taken with various cameras, and we made sure they were out-of-distribution examples of scenes and vantage points not present in MaSTr1325 [2]. Since IMU data (horizon) are not available for these images, we re-trained WaSR${}_{\mathrm{SLR}}$ without IMU on MaSTr1325. The results on the new images are shown in Figure 8.

A remarkable generalization to various conditions not present in the training data are observed. For example, WaSR${}_{\mathrm{SLR}}$ performs well even in certain night-time scenes, even though only daytime images are present in MaSTr1325. Furthermore, the training data contain only open, coastal sea cases, while high performance is obtained even in rivers and tight canals with different water colors and surface textures. We observe failure cases on thin structures (e.g., part of the surfboard missing) and false-positive detections on objects seen through shallow water. We also observe less accurate segmentation of the top-most parts of static obstacles, which is not critical for obstacle detection, but indicates that further research could be invested in maritime segmentation networks to address such cases.

## 5. Conclusions

In this work, we proposed a novel scaffolding learning regime (SLR) for training maritime segmentation models for obstacle detection using only weak, detection-oriented, annotations. The main advantage of SLR is that it focuses only on aspects important for obstacle detection and significantly reduces the required labeling effort (by approximately 20 times). Models trained using SLR do not only match the performance of classically trained models but outperform them by a large margin, setting a new state-of-the-art in segmentation-based maritime obstacle detection. Experiments indicate excellent domain generalization capabilities and a substantial potential in semi-supervised learning and domain adaptation. We also observed that SLR boosts the performance of different network architectures.

In our future work, we plan to explore applications beyond the maritime domain such as autonomous cars and biomedical detection methods based on segmentation. Since SLR is complementary to semi-supervised learning methods such as [7], a combination or fusion of these approaches is another interesting research venue with a wide application spectrum.

## Figures and Tables

**Figure 1 sensors-22-09139-f001:**
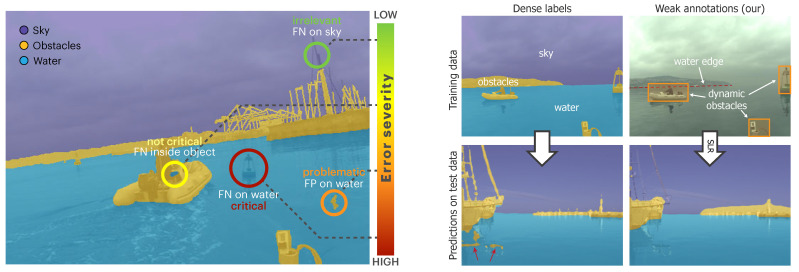
(**Left**) Equal-sized obstacle segmentation errors in different regions have significantly different implications for the downstream task of collision avoidance and boat navigation. (**Right**) The proposed scaffolding learning regime (SLR) trains a network with weak obstacle-oriented annotations (top right) without compromising the segmentation quality relevant to obstacle detection (bottom).

**Figure 2 sensors-22-09139-f002:**
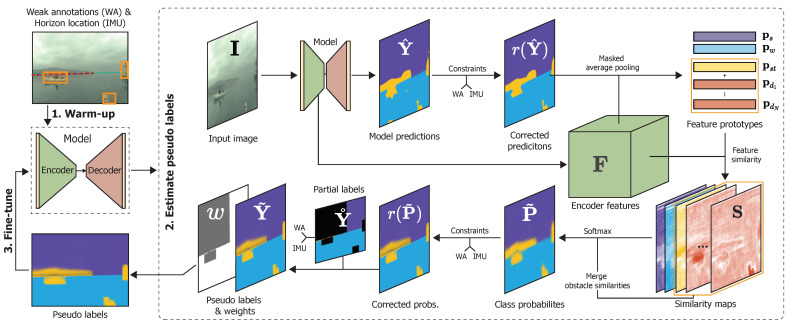
The proposed scaffolding approach SLR is comprised of three steps. (1) The model is warmed-up using object-wise and global objectives derived from weak annotations and IMU. (2) The learned encoder features and model predictions are used to estimate the pseudo labels. (3) The network is fine-tuned with the estimated pseudo labels.

**Figure 3 sensors-22-09139-f003:**
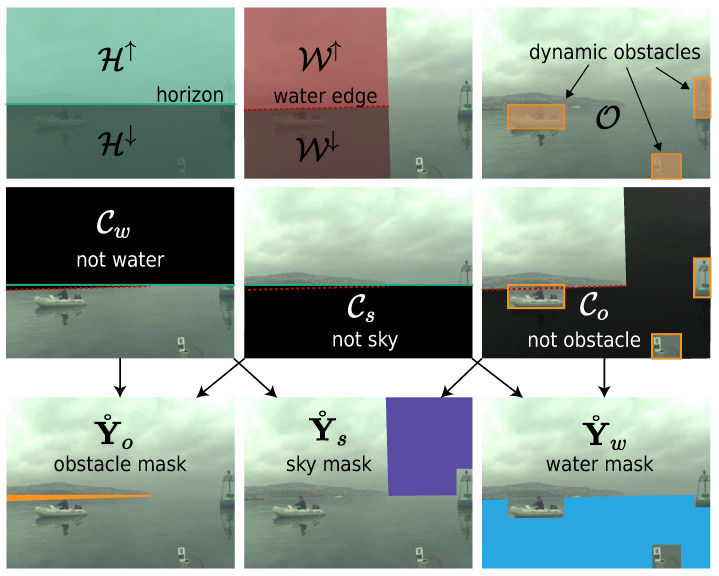
The weak annotations (**top**) form domain-specific constraints (**middle**), which restrict the possible per-pixel labels and generate the initial partial labels (**bottom**).

**Figure 4 sensors-22-09139-f004:**
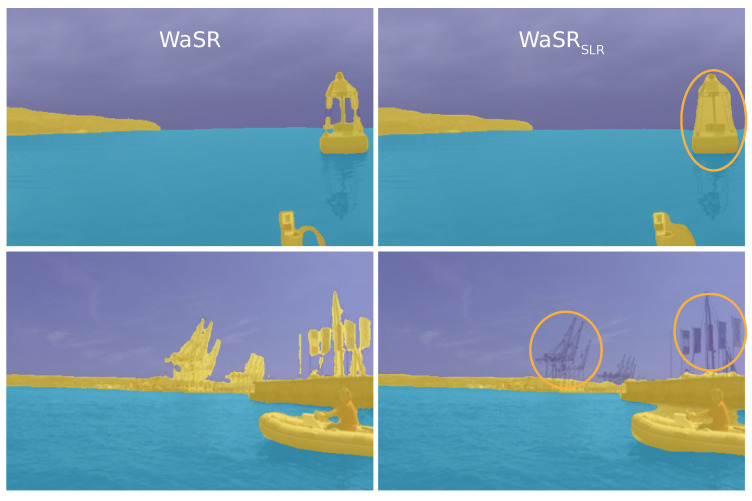
Segmentation obtained by WaSR trained with segmentation ground truth (**left**) and SLR (**right**). SLR sacrifices segmentation accuracy in regions irrelevant for obstacle detection performance (denoted with yellow circles).

**Figure 5 sensors-22-09139-f005:**
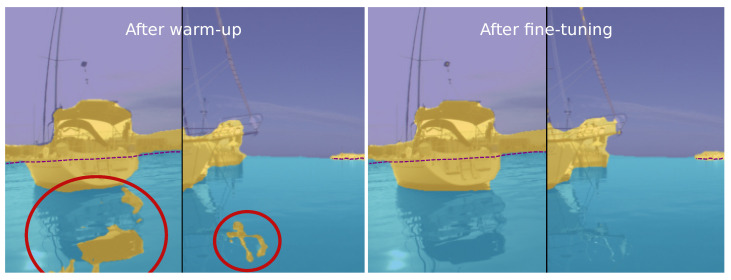
The network predicts false positives (highlighted with red circles) on, e.g., reflections after the warm-up phase (**left**), while performance improves substantially with fine-tuning on re-estimated labels (**right**).

**Figure 6 sensors-22-09139-f006:**
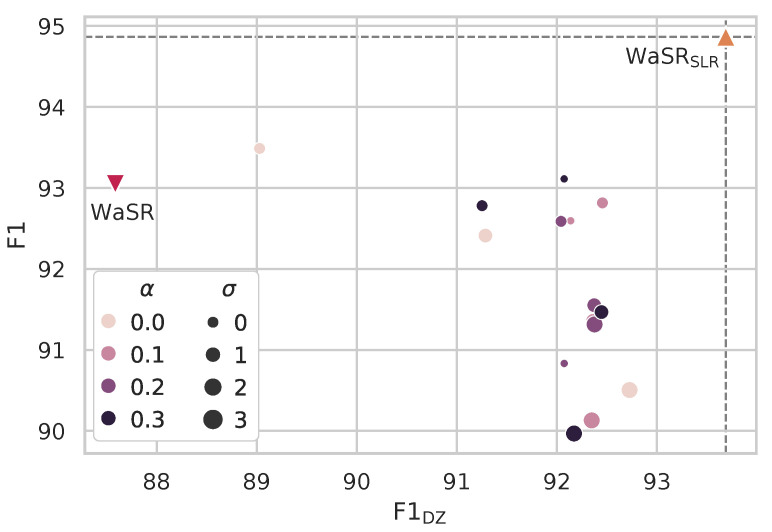
SLR pseudo-label estimation from weak annotations outperforms all combinations of spatial (σ) and pixel-wise (α) label smoothing of dense GT in terms of overall and within danger zone obstacle detection F1 scores.

**Figure 7 sensors-22-09139-f007:**
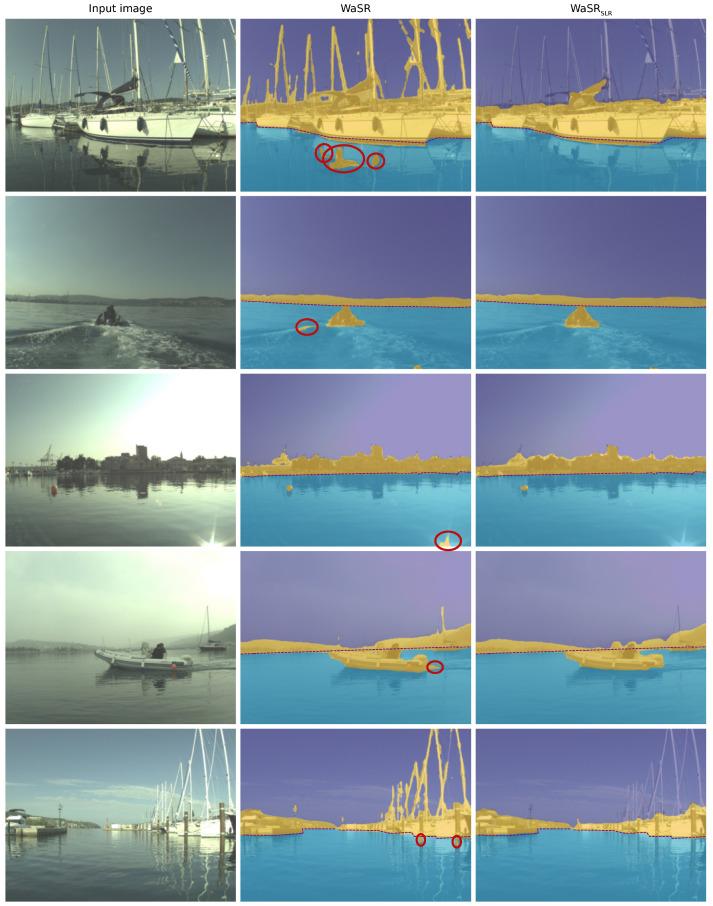
Comparison of WaSR and WaSR${}_{\mathrm{SLR}}$ on the MODS benchmark. Errors are highlighted with red circles. WaSR${}_{\mathrm{SLR}}$ is less sensitive to false-positive detections on object reflections (rows 1 and 5) and other challenging anomalies such as wakes (rows 2 and 4) and sun glitter (row 3).

**Figure 8 sensors-22-09139-f008:**
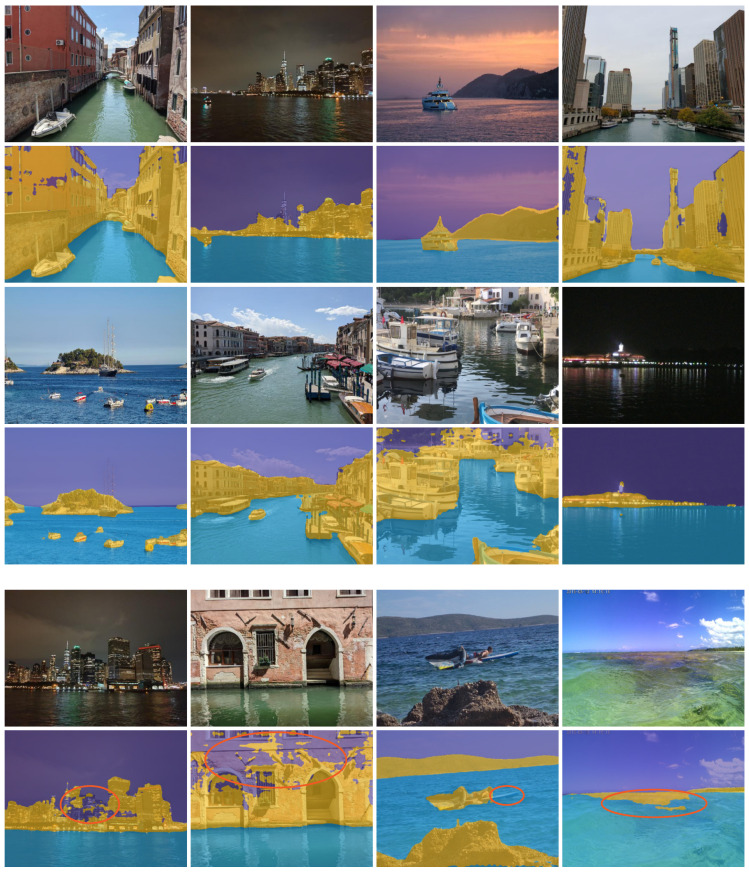
We observe strong performance of WaSR${}_{\mathrm{SLR}}$ in diverse maritime scenarios, including inland waters, different times of day and changes in water color. Failure cases are outlined in the last row and include sky hallucinations, partial detections of thin objects, and false detections of objects visible through the water. Orange circles highlight the areas where the segmentation is not accurate.

**Table 1 sensors-22-09139-t001:** Comparison of methods trained with weak annotations using the proposed SLR training (denoted by ${\left(\cdot \right)}_{\mathrm{SLR}}$) and state-of-the-art methods trained classically with dense ground truth labels on the MODS benchmark [1]. Performance is reported in terms of F1 score, precision (Pr), and recall (Re) for dynamic obstacle detection and water–edge detection robustness ($\mu_{R}$). Best results for each metric are denoted in bold.

Model	$\mu_{R}$	Overall			Danger Zone (<15 m)		
		Pr	Re	F1	Pr	Re	F1
RefineNet	97.3	89.0	93.0	91.0	45.1	98.1	61.8
DeepLabV3	96.8	80.1	92.7	86.0	18.6	**98.4**	31.3
BiSeNet	97.4	90.5	89.9	90.2	53.7	97.0	69.1
WaSR	**97.5**	95.4	91.7	93.5	82.3	96.1	88.6
DeepLabV3${}_{\mathrm{SLR}}$	97.1	94.3	89.4	91.8 (+5.8)	85.5	95.5	90.3 (+59)
WaSR${}_{\mathrm{SLR}}$	97.3	**96.7**	**93.1**	**94.9** (+1.4)	**91.5**	96.0	**93.7** (+5.1)

**Table 2 sensors-22-09139-t002:** Segmentation accuracy (IoU) of fully-supervised WaSR and WaSR${}_{\mathrm{SLR}}$ for the water, sky, and obstacles class, summarized by the mean over classes.

	IoU (Water)	IoU (Sky)	IoU (Obstacles)	mIoU
WaSR	99.7	99.8	98.1	99.2
WaSR${}_{\mathrm{SLR}}$	99.4	99.4	95.0	98.0

**Table 3 sensors-22-09139-t003:** Annotation efficiency of methods (percentage of labelled images, and labeling effort relative to SLR) in terms of obstacle detection performance on MODS (F1 and F1${}_{D}$) and segmentation on the MaSTr1325 validation set (mIoU). Best results for each metric are denoted in bold.

	Labeled Images	Labeling Effort	mIoU (Validation)	F1	F1${}_{D}$
WaSR	5%	1×	96.3	83.8	69.5
	10%	2×	98.4	87.3	78.7
	100%	20×	**99.8**	93.5	88.6
WaSR${}_{\mathrm{ATSO}}$	5%	1×	97.6	90.1	87.5
	10%	2×	98.6	91.4	87.0
WaSR${}_{\mathrm{SLR}}$	100%	1×	98.6	**94.9**	**93.7**

**Table 4 sensors-22-09139-t004:** Domain generalization performance on the SMD test set [30]. Models are trained on MaSTr1325 and the bottom two are additionally fine-tuned on the SMD training set. Best results for each metric are denoted in bold.

	Pr	Re	F1
WaSR	87.3	67.0	75.8
WaSR${}_{\mathrm{SLR}}$	**92.8**	71.7	80.9 (+5.1)
WaSR${}_{\mathrm{FDA}}^{\mathrm{ft}}$	86.9	41.5	56.2 (−19.6)
WaSR${}_{\mathrm{SLR}}^{\mathrm{ft}}$	85.6	**90.2**	**90.7** (+14.9)

**Table 5 sensors-22-09139-t005:** Ablation study on MODS (detection F1/F1${}_{D}$) and MaSTr1325 validation set segmentation (mIoU). Enabling: water–edge static-obstacle heuristic (Equation (Equation 2)), model fine-tuning (Section 3.3), applying constraints r(·) during fine-tuning (Section 3.2), feature clustering for re-estimating pseudo labels (Section 3.2) and the auxiliary warm-up loss $L_{\mathrm{aux}}$ (Section 3.1.2). Best results for each metric are denoted in bold.

Water-Edge Heuristic	Fine-Tuning	Constraints r(·)	Feature Clustering	Auxiliary Loss $L_{\mathrm{aux}}$	mIoU (val)	F1	F1${}_{D}$
					97.4	87.7	54.9
✓					97.4	89.4 (+1.7)	71.0 (+16.1)
✓	✓				97.8	90.5 (+2.8)	84.9 (+30.0)
✓	✓	✓			98.2	93.0 (+5.3)	89.3 (+34.4)
✓	✓		✓		98.5	93.4 (+5.7)	89.7 (+34.8)
✓	✓	✓	✓		98.3	94.2 (+6.5)	91.4 (+36.5)
✓	✓	✓	✓	✓	**98.6**	**94.9** (+7.2)	**93.7** (+38.8)

**Table 6 sensors-22-09139-t006:** MODS detection (F1/F1${}_{D}$) and water–edge estimation robustness ($\mu_{R}$) with respect to the number of SLR pseudo-label estimation and fine-tuning iterations. Best results for each metric are denoted in bold.

Iterations	$\mu_{R}$	F1	F1${}_{D}$
0	**97.8**	87.4	57.5
1	97.3	**94.9**	**93.7**
2	97.0	94.2	92.1
3	96.9	94.3	93.2
4	97.0	93.7	92.5
5	96.9	93.7	92.4

## Data Availability

SLR: Code available at https://github.com/lojzezust/SLR (accessed on 20 October 2022); MaSTr1325: Publicly available at https://www.vicos.si/resources/mastr1325/ (accessed on 20 October 2022); MaSTr1325 Weak Annotations: Publicly available at https://github.com/lojzezust/SLR (accessed on 20 October 2022); MODS: Publicly available at https://github.com/bborja/mods_evaluation (accessed on 20 October 2022).

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
