# Peer review of "Learning with Weak Annotations for Robust Maritime Obstacle Detection"

_sensors, 2022, doi:10.3390/s22239139_

Round 1
Reviewer 1 Report
The manuscript on Learning with Weak Annotations for Robust Maritime Obstacle Detection is well-written in an engaging and lively style.
This topic mentioned in the manuscript is good, and it can be approved for its publication after the necessary comments are addressed.
The authors need to do thorough proofreading of this paper. Some phrases don't make sense and there are lots of typos and wrong use of prepositions.
In the introduction, the findings of the present research work Robust Maritime Obstacle Detection should be compared with the recent work of the same field towards claiming the contribution made.
Authors may refer the recent works like
Multi-modal prediction of breast cancer using particle swarm optimization with non-dominating sorting.
A novel approach for home surveillance system using IoT adaptive security
Authors further need to elaborate on the usage and importance of equations 5 and 7.
It is good to see the outcome of this work on RMSProp optimizer with momentum, but, the authors need to specify how it compares to other existing schemes.
Reviewer 2 Report
The paper presents an interesting subject, but the following aspects must be more clearly explained:
-related work must contain also results
-the previous published paper must be more clearly presented: now it is not clearly what was published and what are the novelties for this paper
-final loss is used only for new method?
-obstacles are detected from images: what should it happen in case of moving obstacles: for example what should it happen for a floating object that appears, disappears, appears and so on
-what is the inference time - is the method suitable for real time applications?
-in case of real life applications, what are the hardware requirements (cameras, suitable distance between the camera and the obstacles in order to be possible to detect the obstacle)
Round 2
Reviewer 2 Report
No comments.